# Perceived Health Benefits in Vestibular Schwannoma Patients with Long-Term Postoperative Headache: Insights from Personality Traits and Pain Coping—A Cross-Sectional Study

**DOI:** 10.3390/jpm14010075

**Published:** 2024-01-08

**Authors:** Mareike Thomas, Hannah Führes, Maximilian Scheer, Stefan Rampp, Christian Strauss, Robby Schönfeld, Bernd Leplow

**Affiliations:** 1Department of Medical Psychology, University Medical Center Hamburg-Eppendorf, Martinistraße 52, 20246 Hamburg, Germany; 2Department of Neurosurgery, University Hospital Halle, Ernst-Grube-Straße 40, 06120 Halle, Germany; 3Department of Neurosurgery, University Hospital Erlangen, Schwabachanlage 6, 91054 Erlangen, Germany; 4Department of Neuroradiology, University Hospital Erlangen, Schwabachanlage 6, 91054 Erlangen, Germany; 5Department of Psychology, Martin-Luther-University Halle-Wittenberg, Emil-Abderhalden-Straße 26-27, 06108 Halle, Germany

**Keywords:** vestibular schwannoma, postoperative headache, microsurgery, personality traits, coping mechanisms, perceived health benefits, pain coping

## Abstract

Postoperative headaches (POHs) following retrosigmoid microsurgery for vestibular schwannoma (VS) can significantly impact patients’ perceived health benefits (PHBs). In this cross-sectional observational study, 101 VS patients were investigated. For the assessment of pain, the Rostock Headache Compendium (RoKoKo) and the German pain processing questionnaire (FESV) were used. The perceived health benefits (PHBs) were assessed by the Glasgow Benefit Inventory (GBI) and Big Five personality traits were measured using the Ten-Item Personality Inventory (TIPI-G). We showed that 55% of the participants experienced POHs, leading to a marked reduction in overall PHBs compared to those without POHs. The correlation analysis revealed an association between decreased PHBs and elevated levels of pain-related helplessness, depression, anxiety, and anger. Positive correlations were identified between PHBs and action-planning competence, cognitive restructuring, and the experience of competence. Low emotional stability and openness yielded associations with pain-related psychological impairment. Hearing loss and facial paresis did not exert a significant impact on PHBs. The study highlights the influence of pain-related coping strategies on PHBs in long-term POH patients. Thus, coping mechanisms and personality traits should be assessed even before surgery for post-surgery pain prevention. The limitations of this study include a relatively small sample size, potential biases introduced by the overrepresentation of female patients, and the use of an online survey methodology. In conclusion, this research highlights that the interplay between headaches, PHBs, and psychological factors is also relevant in VS patients undergoing microsurgery. Short-term psychological interventions should therefore be taken into account to improve post-surgery adaptive coping strategies.

## 1. Introduction

Vestibular schwannoma (VS) is a benign tumor originating from the N. vestibulocochlearis and microsurgery is a common treatment option. While microsurgery is a prevalent choice, studies indicate that it may result in a decrease in perceived health benefits (PHBs) for smaller VSs compared to conservative and radiotherapeutic approaches [1,2,3] However, this negative impact may be reduced over time [4]. Moving beyond treatment modalities, several symptoms affect PHBs in VS patients. These include age [5,6], vertigo [6,7,8,9], impaired facial nerve function [2,5], depression [3,8], anxiety [9,10], and postoperative headaches (POHs) [8,9,11]. POHs have been shown to be especially influenced by surgical approach, age, and tumor size [12]. Studies suggest that the retrosigmoid approach is associated with higher rates of POHs [13,14]. These factors set the stage for exploring the broader psychological landscape in VS patients undergoing microsurgery, such as personality traits.

Patients’ psychological responses to the surgery, their ability to cope with postoperative symptoms, and their overall adjustment to the changes in health status may vary based on their individual personality traits. Exploring these aspects can provide valuable insights into optimizing postoperative care and improving the overall well-being of VS patients undergoing microsurgery.

In our initial project investigation, we examined links between premorbid psychological factors and POHs, discovering a higher prevalence of premorbid mental ailments, preexisting headaches, chronic pain syndromes, and elevated somatization tendencies in those with POHs. Regression analyses indicated that premorbid psychosomatic symptoms, mental ailments, and chronic pain syndromes predicted POHs, highlighting the importance of psychological screening before microsurgery [15]. Despite limited literature on psychological factors in VS patients, Ribeyre et al. [16] emphasize that psychological traits significantly contribute to postural recovery after surgery-related balance impairment after VS surgery. Traits such as high anxiety, focus on negative emotions, and challenges in self-care may hinder the recovery process. This transition sets the groundwork for understanding the psychological aspects beyond the physical symptoms.

Given that VS patients who undergo retrosigmoid surgery frequently experience POHs [13,14], there is a natural progression to exploring associations between headache types and personality traits. Several studies shed light on the complex relationship between personality traits and primary headaches. In a meta-analysis conducted by Garramone et al. [17], it was observed that migraineurs tended to exhibit high neuroticism and low extraversion. Özdemir et al. [18] found that individuals with migraine commonly display maladaptive and ineffective coping responses. Similarly, Aaseth et al. [19] reported that patients with chronic tension-type headaches have significantly elevated neuroticism scores and higher levels of psychological distress compared to the general population. Further, a systematic review [20] identified a “neurotic profile” in individuals experiencing chronic headaches. The analysis demonstrated elevated levels of depressive and anxious personality dimensions, as well as heightened pain in chronic headache patients compared to healthy controls.

PHBs in VS patients may be impacted by psychiatric comorbidities, such as depression [3,8] or anxiety [9,10]. Headaches are also linked to psychiatric comorbidities. Rausa et al. [21] study revealed that 46.8% of chronic daily headache patients (*N* = 94) had psychiatric comorbidities, predominantly mood and anxiety disorders. These patients exhibited elevated scores in the neurotic triad, particularly in hypochondria, along with mild levels of depression and hysteria according to the aequivalent subscales of the MMPI-2. Notably, these patients expressed strong concerns about their health. Additionally, Yang et al. [22] found that comorbidity with personality disorders was associated with more severe forms of migraine symptoms, with migraineurs having an additional diagnosis of personality disorder exhibiting a higher frequency of headaches. Taken together, these studies propose that personality traits may play a pivotal role in shaping how individuals cope with headaches and perceive their overall health benefits.

However, there is still a lack of data regarding associations between personality traits, coping mechanisms, and PHBs in VS patients experiencing long-term POHs. Therefore, the objectives of the current cross-sectional study were to (1) assess the PHBs in individuals with VS experiencing POHs after receiving retrosigmoid microsurgery in comparison to those without POHs; (2) analyze the personality traits of VS patients with long-term POHs; and (3) investigate the associations between personality traits and coping mechanisms concerning POHs in VS patients. Finally, we aimed to identify pain-related psychological impairment and coping mechanisms regarding POHs that could be potential predictors of PHBs.

## 2. Materials and Methods

This cross-sectional, single-center study was part of a larger study carried out at University Hospital Halle, Germany. To enhance the stability of the selected samples in this study, several methodological considerations and adjustments were implemented. The inclusion and exclusion criteria were carefully defined to ensure homogeneity within the participant group, with a focus on patients diagnosed with VS who underwent retrosigmoid surgery, were native German speakers, and were aged 18 or older at diagnosis. Patients with previous surgery or radiation and/or recurrent VS and those with additional oncological diagnosis or neurofibromatosis type 2 were excluded from study participation. We used standardized questionnaires. With the study starting in early 2020, patient recruitment was delayed by the COVID-19 pandemic. Participants were approached directly during the chief resident consultation at University Hospital Halle. After the national lockdown began, the Vereinigung Akustikus Neurinom e.V. (a non-profit patient self-help organization) called for participation to complete an online survey (SoSciSurvey). Both surveys used identical questions. Before participating in the study, all participants provided written consent. The study was approved by the Ethics Committee of University Hospital Halle (No. 2020-008).

A brief self-administered questionnaire was used to determine demographic factors such as age at onset, time of surgery, and gender assigned at birth. Tumor size was assessed as a Koos grade, ranging from 1 to 4 [23]. A POH was characterized as a persistent, ongoing headache that began more than three months following retrosigmoid surgery. The Rostock Headache Compendium (RoKoKo) was used to assess POHs to categorize the symptoms as migraine (continuous), tension-type headaches, or other headaches [24]. The comprehensive findings regarding the POHs are discussed in Thomas et al. [15]. Participants were asked to rank their pain on a numeric analogue scale (NAS) from 1 to 10.

In addition, the patients were evaluated using the Glasgow Benefit Inventory (GBI). The GBI is a validated post-interventional questionnaire consisting of 18 items. It can be used to assess patients’ PHBs after undergoing an intervention, e.g., surgery. The set of 18 questions focuses on various aspects of general, social, and physical health. Scores span from −100 to +100, where a score of zero signifies no benefit, +100 represents the highest level of benefit, and negative scores indicate a worsening of health [25].

In patients with a chronic or recurrent pain issue, the German pain processing questionnaire (Fragebogen zur Erfassung des Schmerzverhaltens, FESV) [26] was used to evaluate the patients’ psychological pain-related mental interference (questionnaire BE) as well as their cognitive and behavioral coping mechanisms regarding pain (questionnaire BW). Three main components and nine individual dimensions were evaluated for pain processing: cognitive pain management, behavioral pain management, and pain-related mental interference. The components of action-planning competence (APC), cognitive restructuring (CR), and experience of competence (EC) constituted the cognitive pain coping assessment. Mental distraction (MD), counteractive activities (CA), and rest and relaxation (RR) strategies are all parts of behavioral pain coping. The subscales of pain-related helplessness and depression (HD), pain-related anxiety (ANX), and pain-related anger (ANG) constituted the pain-related psychological impairment scale [26].

To assess the Big Five personality traits, as described by McCrae and Costa [27], we used the Ten-Item Personality Inventory (TIPI-G) [28]. The five components of this personality model are emotional stability (the polar opposite of neuroticism), extraversion, agreeableness, openness, and conscientiousness. When testing time is restricted, this short version of the questionnaire provides an accurate approximation for longer measurements of the five-factor model of personality, such as the NEO-PI-R [29].

Statistical analyses were performed using SPSS software, version 28.0 [30]. A level of significance of α = 0.05 was applied. Mann–Whitney-U-tests were used for group comparisons. Spearman-Rho (*r_s_*) correlations were calculated for a robust estimation even in the case of a non-normal distribution. False discovery rate correction [31] was applied to account for multiple testing [32]. A step-wise regression analysis was conducted to identify pain-related psychological impairment and coping mechanisms as predictors for PHBs. Goodness-of-fit was determined according to Cohen [33].

## 3. Results

### 3.1. Demographic Characteristics

A total of 101 participants who received microsurgery via the retrosigmoid approach were surveyed, and 50 reported POHs. After excluding 9 patients with incomplete questionnaires or missing tumor sizes, 92 participants were included in the evaluation. For POH patients, the mean age at the time of the survey was 53.9 years (*SD* = 12.7), and the mean age at diagnosis was 45.9 years (*SD* = 10.0). The average time span between surgery and the surgery was 8.0 years (*SD* = 9.4, Median = 3.5). A total of 7 patients reported a Koos grade of 1 (14%), 14 reported a grade of 2 (28%), 20 reported a grade of 3 (40%), and 9 reported a grade of 4 (18%). Thirty-seven participants (74%) were female. The current average POH pain level was ranked at 6.7 (range 1–10). PHB scores, represented by all GBI scores except the social support subscale, exhibited negative means, indicating a reduction in perceived health benefits. Additionally, pain-related mental interference and pain coping demonstrated generally moderate levels. Conscientiousness levels were notably high, whereas emotional stability and extraversion exhibited more moderate levels. The characteristics of the study sample are summarized in Table 1.

### 3.2. Associations between POH and PHB

The participants with POHs reported significantly lower levels of total PHBs (Table 2). The effect size of *d* = 0.52 can be interpreted as medium according to Cohen (33).

### 3.3. Associations between Age, Hearing Loss, Facial Paresis, and Pain

A younger age was associated with higher levels of pain-related anxiety and anger, whereas an older age was associated with higher levels of experience of competence. Hearing loss was associated with older age, high levels of pain-related anxiety and anger, as well as lower levels of cognitive restructuring. With regard to personality traits, hearing loss was associated with low levels of extraversion. There were no significant correlations regarding facial paresis (Appendix A). Pain severity, as reported via NAS, was negatively associated with extraversion, emotional stability, and openness. Age, hearing loss, and facial paresis were not significantly associated with any of the GBI scores. Time since treatment was not significantly associated with general (*r_s_* = 0.05, *p* = 0.76), social support (*r_s_* = 0.20 *p* = 0.16), physical health (*r_s_* = 0.15, *p* = 0.31), and total GBI scores (*r_s_* = 0.20, *p* = 0.17).

### 3.4. Associations between Pain-Related Mental Interference, Pain Coping, and Personality Traits

Figure 1 shows the results of the correlation analysis. High pain-related anxiety and anger levels were associated with low levels of extraversion, emotional stability, and openness. Low emotional stability and openness were also associated with high pain-related helplessness and depression. The experience of competence was significantly associated with extraversion and emotional stability (Appendix A).

### 3.5. Associations between Pain-Related Mental Interference, Coping Mechanisms, and PHBs

Pain-related helplessness and depression, and anxiety were significantly associated with lower general and physical health and total GBI scores. Pain-related anger was also associated with lower total GBI scores (Appendix A). In terms of pain coping, higher levels of action-planning competence, cognitive restructuring, and experience of competence were associated with higher general and total GBI scores. Cognitive restructuring and experience of competence also correlated significantly with the physical health subscale (Figure 2).

### 3.6. Associations between Personality and PHBs

Extraversion was the only personality trait that yielded a significant correlation with PHBs and was associated with the physical health GBI score (Appendix A).

### 3.7. Predictors for PHBs

The step-wise regression analysis revealed certain predictors for the GBI total score, which serves as an indicator of PHBs. Pain-related helplessness and depression were identified as negative predictors (β = −0.34), suggesting that higher levels of pain-related helplessness and depression are associated with lower perceived health benefits. On the other hand, cognitive restructuring emerged as a positive predictor (β = 0.37), implying that individuals who engage in cognitive restructuring tended to report higher perceived health benefits. The overall model’s statistical significance (*F* (2, 46) = 11.13, *p* < 0.01, R2 = 0.33) suggests that these predictors collectively explain a significant portion (33%) of the variability in total GBI scores.

## 4. Discussion

### 4.1. Summary of Findings

In this cross-sectional study, we examined the interplay between personality traits, pain coping mechanisms, and PHBs among VS patients who underwent retrosigmoid microsurgery and experienced POHs. Our study yielded significant insights, revealing associations between personality traits, pain coping, psychological distress, and PHBs. Our first investigation showed that, prior to the surgical procedure, POH patients had already encountered heightened psychological stress during the preoperative phase [15]. Expanding upon this analysis, we have delved into the contemporary associations among enduring personality traits, coping strategies, and post-surgery PHBs.

### 4.2. Associations among Age, Hearing Loss, and Pain-Related Mental Interference

Higher pain-related anxiety and anger were linked to a younger age, while an older age was notably associated with enhanced experienced competence. This could be attributed to the accumulation of pain-coping experience with advancing age, potentially making younger patients more prone to pain-related mental interference due to POHs. Hearing loss was significantly associated with pain-related anxiety and anger. In accordance, the findings of Garnefski and Kraaij, who conducted a cross-sectional survey of patients suffering from hearing loss [34], indicated a connection between ruminative and catastrophizing coping strategies and higher reports of depression and/or anxiety symptoms among individuals with hearing loss. In contrast, refocusing attention to more pleasant issues, disengaging from unattainable goals, and re-engaging in alternative, meaningful goals were associated with less symptomatology. Nevertheless, the impact of POHs appears to be uncertain. The negative association with extraversion may be attributed to communication breakdowns, implying that the impairment is likely to affect frequent communication partners. This discussion contends that the influence of the impairment on others constitutes a concept referred to as “Third-Party Disability” [35]. Contrary to Bender et al. [3], neither hearing loss nor facial paresis showed associations with any of the GBI scores. In fact, facial paresis showed no significant correlations with any of the measures used. In contrast to the findings of Turel et al. [4], who conducted a prospective study to investigate health-related quality of life in patients with large VSs before and after surgery, the time since treatment was not significantly associated with GBI scores.

### 4.3. Personality Traits and Pain-Related Mental Interference and Coping

Regarding personality traits, pain-related mental interference exhibited a negative correlation with extraversion, emotional stability, and openness. This finding shows that individuals with POHs show personality traits similar to those of migraineurs [17,18,19,20]. Magyar et al. [36] investigated the Big Five personality traits, headaches, and lifetime depression in a large European general population sample. They found that openness increased the risk of migraine. They also demonstrated that individuals with migraine but without depression achieved higher scores in openness compared to those experiencing depression. In the context of migraines, individuals with migraines who possess greater levels of openness tended to exhibit enhanced flexibility and creativity in their strategies for managing their condition. As a result, this adaptive strategy helps to alleviate the influence of migraines on their daily routines, as demonstrated by a reduced level of functional disruption in migraine sufferers with elevated openness levels [37]. Nevertheless, our results did not reveal a noteworthy connection between openness and the utilization of adaptive pain-coping strategies. Extraversion and emotional stability were positively associated with experienced competency. These results are in line with Ramírez-Maestre et al. [38], who analyzed the relationships between personality and the intensity of perceived pain and the coping strategies used in a sample of chronic pain patients and found that extraversion was associated with lower levels of pain intensity (NAS).

The theoretical explanations for associations between personality and pain coping often center on the psychobiological aspects of pain perception. Personality traits might influence neurobiological processes, including stress response systems and pain modulation pathways. For example, individuals with high extraversion may exhibit more effective stress-coping mechanisms, thereby mitigating the impact of stress-induced hyperalgesia. Emotional stability, on the other hand, could contribute to a more adaptive pain appraisal, reducing the emotional amplification of pain signals. Moreover, openness as a personality trait may play a role in shaping cognitive and behavioral responses to pain. Open individuals, characterized by creativity and a willingness to experience novel ideas, may adopt more diverse and flexible coping strategies. This adaptability could extend to their ability to reframe pain-related cognitions, fostering a more positive outlook and mitigating mental interference.

### 4.4. Biopsychosocial Model and Diathesis-Stress Component

The biopsychosocial model considers the interconnectedness of the mind and body, addressing biological, psychological, and social aspects of pain and illness [39]. In contrast, the biomedical disease model focuses on bodily disruptions caused by physiological factors. Expanding the biopsychosocial model with a diathesis-stress component could improve chronic pain treatment. The initial distress post-surgery can lead to psychological issues and physical and mental deconditioning, including learned helplessness, anxiety disorders, personality disorders, and unhealthy coping mechanisms. As the pain becomes chronic, assessing psychosocial factors becomes crucial for acceptance, maintenance, and suffering. The emotional effects of pain involve behaviors like catastrophizing, fear avoidance, re-evaluation of beliefs, efficacy, control, vulnerability, and resilience. Catastrophizing and avoidance stem from anticipated pain, exacerbating these behaviors. Patient differences in persistent personality traits as well as self-control and self-efficacy may impact learned helplessness and similar behaviors, but this remains uncertain [40]. Negative affect is likely the most frequently evaluated psychological aspect in individuals with persistent pain. Additionally, emotional strain and psychosocial tension have demonstrated a propensity to escalate the chances of transitioning from acute to chronic musculoskeletal pain [41]. Our results align with the idea that individuals exhibiting specific personality traits, including reduced extraversion, emotional stability, and openness, are more susceptible to the development of POHs following microsurgery.

### 4.5. PHBs and Coping Mechanisms

Regarding PHBs, our findings align with previous studies [8,9,11], indicating that individuals with POHs have significantly lower total GBI scores (PHB) compared to those without POHs. In individuals experiencing POHs, pain-related helplessness, depression, and anxiety exhibited negative correlations with the general, physical health, and overall scores of the GBI. Additionally, elevated levels of pain-related anger were linked to reduced overall GBI scores. These findings suggest that patients experiencing greater pain-related interference regarding POHs tended to report notably lower levels of PHBs following microsurgery. Regarding coping mechanisms, elevated levels of action-planning competence, cognitive restructuring, and experienced competence were indicative of heightened general, physical health, and overall scores within the GBI assessment. Furthermore, action-planning competence scores exhibited a positive correlation with increased physical health scores in the GBI. Scheidegger et al. [41] analyzed 602 chronic primary pain inpatients and revealed three subtypes: (1) severely burdened individuals with low coping skills, (2) mildly burdened individuals with high coping skills, and (3) moderately burdened individuals with moderate coping skills. All subtypes showed improved pain interference, psychological distress, and coping skills after treatment. Pain-related mental interference improved notably in subtypes (1) and (3), while pain intensity decreased significantly only in subtype (3). For subtype (1), targeting relaxation techniques, counteractive activities, and cognitive restructuring post-treatment seemed to be the most promising approached for reducing pain interference and psychological distress. FESV dimensions did not significantly predict treatment outcomes in subtype (2). Subtype (3) individuals could gain the most by enhancing their sense of competence during treatment. Their results were in accordance with Grolimund et al. [42]. Future research on POHs following VS removal should explore various coping subtypes to determine the most effective treatment approaches.

### 4.6. Psychological Interventions for POHs in VS Patients

Psychological interventions for individuals dealing with POHs could be a suitable approach to enhance PHBs following microsurgical removal of VSs. Following treatment, patients with chronic primary pain experienced notable enhancements in symptoms, cognitive pain coping, and behavioral pain coping. Specifically, distinct improvements were observed in both cognitive and behavioral coping skills. However, when examined through hierarchical linear models, pain coping did not show significant links with decreases in pain intensity. On the other hand, the overall level and enhancements in cognitive pain coping were predictive of reductions in pain interference and psychological distress. From a clinical perspective, it could prove valuable to encourage and engage in cognitive restructuring and action planning during treatment to effectively diminish post-treatment levels of pain interference and psychological distress [43]. Employing strategies to reduce pain-related helplessness and depression, alongside fostering cognitive restructuring skills, appears to be a promising avenue for treatment of POHs, as indicated by the regression analysis.

### 4.7. Limitations

It is important to consider the study’s results in light of certain methodological limitations. Firstly, a potential limitation of this study is the relatively small sample size of 54 VS patients experiencing POHs. With a limited number of participants, the findings may not fully capture the diversity and complexity of the population under investigation. A reduced sample size could compromise the statistical power, potentially impeding the identification of subtle or less prevalent effects and associations. While this study offers valuable insights, caution is warranted when extrapolating the results to a larger population of VS patients, highlighting the need for prudence in drawing broad conclusions. Nevertheless, initiating investigations with a modest sample size is a judicious approach, preventing undue resource allocation in the absence of established associations between factors and disorders. This strategy helps prevent excessive allocation of resources such as subjects, time, and financial investments towards establishing an association between a factor and a disorder, especially in cases where no actual effect exists [44].

Another noteworthy consideration is the notable overrepresentation of female patients in the study, potentially introducing a gender-related bias that might impact the generalizability of the findings. The predominance of one gender raises concerns about the applicability of the results to a more diverse population, as responses and outcomes may differ between males and females. The sex imbalance could potentially skew associations or place undue emphasis on factors that predominantly affect females. However, it is essential to note that the statistical analysis did not reveal significant effects related to sex at birth.

Employing online surveys introduces drawbacks, as the absence of an interviewer precludes the opportunity to address unfamiliar or ambiguous terms. Additionally, the use of an online survey methodology introduces potential limitations related to data accuracy. The reliance on self-reported data via online surveys may be subject to recall bias and participant interpretation, potentially influencing the reliability of the responses. The absence of direct researcher oversight in an online setting may raise concerns about the consistency and accuracy of the participant responses. Furthermore, this approach may not capture responses from individuals without internet access, particularly elderly patients, potentially leading to a bias in the respondent pool. Additionally, the anonymity of participation could increase the risk of fraudulent responses [45].

## 5. Conclusions

In conclusion, this study sheds light on the significant impact of POHs on the PHBs of patients who undergo retrosigmoid microsurgery for VS. Our findings indicate that patients experiencing long-term POHs exhibit a notable reduction in overall PHBs compared to those without POHs. The link between PHBs and personality traits, pain-related mental interference, and pain coping is evident, underscoring the intricate interplay between these factors. Specifically, our study underscores that heightened levels of pain-related helplessness, depression, anxiety, and anger are closely associated with lower PHBs among POH patients. Conversely, psychological skills such as action-planning competence, cognitive restructuring, and experience of competence demonstrate a positive correlation with PHBs. Notably, personality traits, particularly extraversion, emotional stability, and openness, are implicated in the extent of pain-related mental interference.

Remarkably, the presence of hearing loss and facial paresis was not found to influence PHBs. This suggests that the psychological and emotional dimensions of pain and coping might be more influential determinants in shaping patients’ perceived health benefits post-surgery.

These insights emphasize the importance of addressing pain-related mental interference and implementing effective coping mechanisms to enhance PHBs in VS patients grappling with long-term POHs. Consequently, these insights also unveil implicit implications for preoperative patient education, specifically within the context of informed consent. Interventions focusing on cognitive restructuring hold promise in fostering adaptive coping strategies among POH patients. These findings pave the way for future research and the development of interventions aimed at optimizing the postoperative experience and overall well-being of VS patients enduring the challenges of POHs following microsurgery.

Future studies with larger and more balanced and representative samples of both male and female patients could help validate and strengthen the observed relationships and contribute to a more comprehensive understanding of the factors influencing postoperative outcomes in this context. Conducting longitudinal studies would provide valuable insights into the dynamic nature of postoperative outcomes over time. Tracking patients at multiple points post-surgery can elucidate the trajectory of symptoms, coping mechanisms, and psychological well-being, offering a more comprehensive understanding of the long-term impact of surgery on VS patients. Beyond personality traits, researchers could explore additional factors that may influence postoperative outcomes. This may include examining the role of social support, coping strategies, and specific cognitive-behavioral factors. Understanding the interplay between these variables and their collective impact on postoperative well-being could guide the development of targeted interventions. Lastly, collaborative efforts between neurosurgeons, psychologists, and other healthcare professionals could foster a holistic understanding of postoperative outcomes. Multidisciplinary studies could explore how a combination of medical, psychological, and supportive interventions contributes to better patient outcomes.

## Figures and Tables

**Figure 1 jpm-14-00075-f001:**
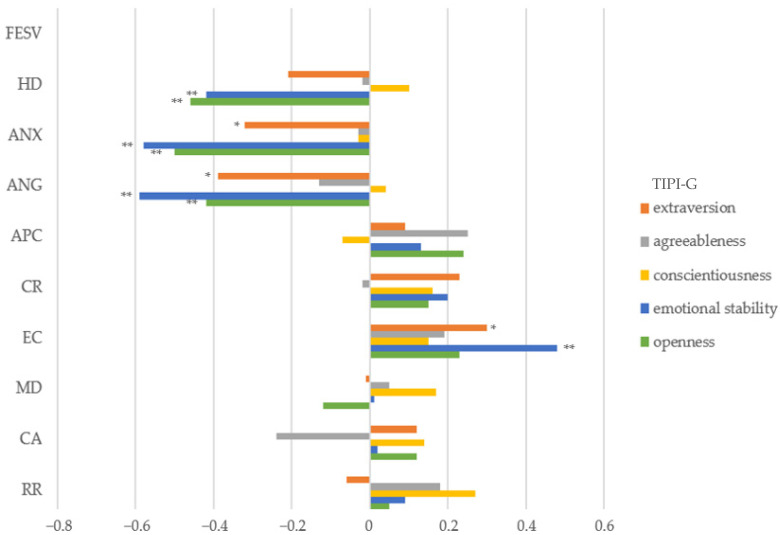
Correlation analysis for pain-related mental interference, pain coping, and personality traits. Abbreviations: TIPI-G = Ten-Item Personality Inventory German, FESV = questionnaire for the assessment of pain-related behavior, HD = helplessness and depression, ANX = anxiety, ANG = anger, APC = action-planning competence, CR = cognitive restructuring, EC = experience of competence, MD = mental distraction, CA = counteractive activities, RR = rest and relaxation. * *p* < 0.05, ** *p* < 0.001.

**Figure 2 jpm-14-00075-f002:**
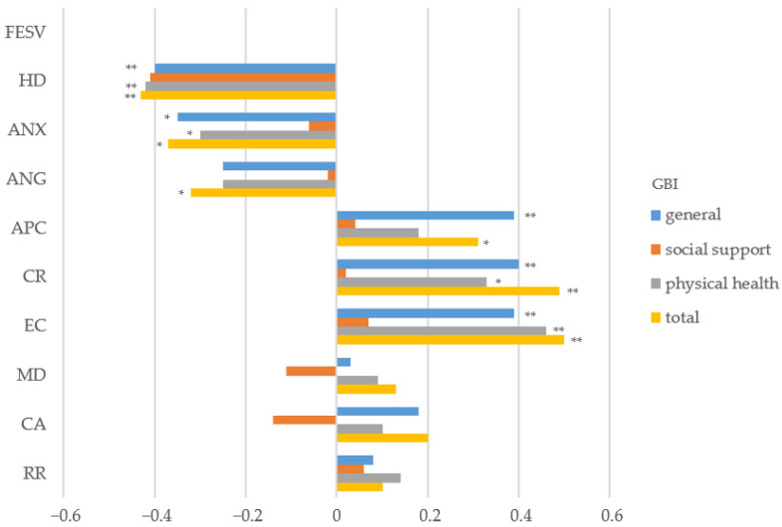
Correlation analysis for pain-related mental interference, pain coping, and PHBs. Abbreviations: TIPI-G = Ten-Item Personality Inventory German, FESV = questionnaire for the assessment of pain-related behavior, HD = helplessness and depression, ANX = anxiety, ANG = anger, APC = action-planning competence, CR = cognitive restructuring, EC = experience of competence, MD = mental distraction, CA = counteractive activities, RR = rest and relaxation., GBI = Glasgow Benefit Inventory. * *p* < 0.05, ** *p* < 0.001.

**Table 1 jpm-14-00075-t001:** Descriptive data of demographic variables and questionnaires responses of VS patients experiencing POHs (*n* = 50).

Variable/ Questionnaire Scale	Minimum	Maximum	M (SD)
Age	30	75	53.94 (12.68)
Age at onset	28	69	45.92 (10.04)
Time since treatment in years	1	36	8.02 (9.36)
Pain (NAS)	2	10	6.65 (2.13)
TIPI-G			
Extraversion	2	14	8.04 (3.20)
Agreeableness	6	14	10.50 (2.23)
Conscientiousness	8	14	12.37 (1.58)
Emotional stability	2	14	9.6 (3.34)
Openness	5	14	10.58 (2.48)
FESV			
HD	5	30	15.40 (7.71)
ANX	4	24	10.77 (6.12)
ANG	5	26	11.15 (6.29)
APC	4	24	15.55 (5.33)
CR	4	24	14.70 (5.56)
EC	4	24	16.08 (5.56)
MD	4	18	7.94 (4.01)
CA	3	20	8.34 (4.30)
RR	4	24	11.49 (5.58)
GBI			
Global	−100.00	75.00	−23.19 (34.93)
Social support	−83.33	66.67	0.31 (27.44)
Physical health	−83.33	33.33	−21.96 (31.86)
Overall	−80.56	63.69	−20.09 (25.68)

Abbreviations: TIPI-G = Ten-Item Personality Inventory German, FESV = German pain processing questionnaire, NAS = numerical analogue scale, HD = helplessness and depression, ANX = anxiety, ANG = anger, APC = action-planning competence, CR = cognitive restructuring, EC = experience of competence, MD = mental distraction, CA = counteractive activities, RR = rest and relaxation, GBI = Global Benefit Inventory.

**Table 2 jpm-14-00075-t002:** Mann–Whitney *U*-tests for group comparisons of patients with and without POHs regarding PHBs.

Questionnaire Scale	With POHs (*n* = 50)	Without POHs (*n* = 42)	*p*	FDR-Corrected *p*	Cohen’s *d*
GBI					
General	−23.4 (34.6)	−10.9 (27.5)	0.16	0.21	-
Social support	−0.9 (29.5)	1.1 (20.6)	0.56	0.56	-
Physical health	−23.2 (32.0)	−9.1 (21.5)	0.049	0.098	-
Total	−20.7 (25.3)	−8.6 (19.9)	0.01	0.04	0.52

Abbreviations: FDR = false detection rate, GBI = Global Benefit Inventory.

## Data Availability

The data presented in this study are available on request from the corresponding author. The data are not publicly available due to privacy reasons.

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
