# Peer review of "Perceived Health Benefits in Vestibular Schwannoma Patients with Long-Term Postoperative Headache: Insights from Personality Traits and Pain Coping—A Cross-Sectional Study"

_jpm, 2024, doi:10.3390/jpm14010075_

Round 1

Reviewer 1 Report

Comments and Suggestions for Authors

1.       Some sentences are lengthy and might benefit from being broken down for clarity. For example, the sentence starting with "A survey of 17 101 VS patients..." is quite long and could be rephrased for better readability.

2.       The abstract mentions "almost half" of the patients experiencing postoperative headaches, but it might be helpful to provide a specific percentage for a clearer understanding.

3.       While the abstract mentions the use of various surveys and questionnaires, it does not provide details on the methodology used for the study. Including a brief overview of the study design and methods could enhance the abstract.

4.       The abstract discusses associations between postoperative headaches and perceived health benefits, but it does not explicitly mention the direction of causality. Clarifying whether postoperative headaches directly lead to reduced perceived health benefits or if there are other factors at play could strengthen the abstract.

5.       The abstract briefly mentions short-term psychological interventions, but it could benefit from more explicit recommendations for clinical practice or further research based on the study's findings.

6.        It's important to acknowledge any limitations in the study. For example, potential biases in survey responses, generalizability of findings, or other methodological constraints should be briefly mentioned.

7.       The abstract could conclude with a concise summary of the key findings and their potential implications, reinforcing the significance of the study

8.       The introduction covers a range of topics, from the impact of microsurgery on perceived health benefits (PHB) to the association between personality traits and postural recovery. Consider restructuring the introduction to present a clearer flow of ideas, possibly by dividing it into subsections that focus on specific aspects of the study.

9.       While the objectives of the study are mentioned towards the end of the introduction, it might be helpful to explicitly state them earlier for the reader to understand the purpose of the study right from the beginning.

10.   The introduction refers to several previous studies, which is good for contextualizing the research. However, it would be more informative if specific citations were provided for each referenced study, allowing readers to access the relevant literature.

11.   Some transitions between sentences and ideas could be smoother. Consider using transition phrases to guide the reader through the logical progression of information.

12.   The section on personality traits and their impact on postural recovery is informative but could benefit from a brief explanation of how these traits are relevant to the study of vestibular schwannoma (VS) patients.

13.   While it's mentioned that microsurgery might lead to a decrease in PHB compared to conservative therapy, it would be helpful to briefly discuss why this is the case and how it relates to the current study.

14.   Ensure consistency in the terminology

15.   While the descriptive data in Table 1 is useful, consider incorporating more narrative explanation to highlight key findings before directing readers to the tables. For instance, briefly discuss notable trends or patterns in demographic variables before presenting the detailed table.

16.   The description of the study sample includes both "50" and "54" participants with POH, and later in the results section, "50" is consistently used. Clarify the reason for this discrepancy and ensure consistency in reporting.

17.   The correlation matrix (Table 3) is extensive and includes various variables. Consider providing a brief narrative interpretation of the most relevant or significant correlations before referring readers to the table for detailed information. This helps readers understand the key relationships without needing to analyze the entire matrix.

18.   In Table 2, Cohen's d is provided as a measure of effect size for the Mann-Whitney-U-Tests. Consider providing a brief interpretation or reference point for the Cohen's d values to help readers understand the practical significance of the observed differences.

19.   Ensure that all abbreviations used in the tables (e.g., NAS, TIPI-G, FESV, HD, ANX, ANG, APC, CR, EC, MD, CA, RR, GBI) are defined either in the text or in a footnote to enhance reader understanding.

20.   To enhance the structure and readability, consider using subsection headings for different aspects of the results (e.g., "3.1. Demographic Characteristics," "3.2. Associations between Age, Hearing Loss, Facial Paresis, and Pain").

21.   In the last part of the results, where predictors for perceived health benefits are discussed, provide more interpretation of the findings. Explain the significance of pain-related helplessness and depression and cognitive restructuring as predictors and their implications for perceived health benefits.

22.   Depending on the complexity of the data, consider incorporating visual aids such as graphs or charts to help readers grasp key trends or relationships more easily.

23.   he discussion covers a wide range of topics, including associations among personality traits, pain coping, psychological distress, and perceived health benefits (PHB). Consider organizing the discussion into subsections to enhance clarity. For example, have separate sections for personality traits, pain coping, psychological distress, and PHB.

24.   While the discussion provides a detailed summary of the findings, consider delving deeper into the potential mechanisms or reasons behind the observed associations. For instance, when discussing the association between personality traits and pain-related mental interference, explore theoretical explanations or existing literature supporting these associations.

25.   When referring to previous studies (e.g., Garnefski and Kraaij, Turel et al., Magyar et al., Ramírez-Maestre et al.), provide more context and specifics about the methodologies and results of those studies. This will help readers understand the relevance and significance of the current findings in the context of existing literature.

26.   The limitations section is appropriately included, but it might benefit from further discussion. For example, discuss the potential impact of the sample size on the study's generalizability and the potential biases introduced by the overrepresentation of female patients. Additionally, acknowledge any potential limitations related to the online survey methodology and its implications for data accuracy.

27.   The discussion briefly touches on the clinical implications of the findings, particularly in terms of preoperative patient education and interventions. Elaborate on these implications, discussing how the study results could inform clinical practices, patient care, and potential interventions for patients experiencing postoperative hearing-related issues.

28.   The conclusion briefly mentions future research with larger and more balanced samples. Provide more specific recommendations for future research directions, such as exploring additional factors that may influence postoperative outcomes, investigating the effectiveness of specific interventions, or considering longitudinal studies to assess changes over time.

29.   Ensure that each statement, especially those making comparisons with previous studies or presenting theoretical concepts, is supported by appropriate citations. This enhances the credibility of the discussion and provides readers with the opportunity to explore relevant literature.

Comments on the Quality of English Language

minor editing required

Author Response

Dear Reviewer,

thank you very much for investing the time to review our manuscript. We really appreciate your valuable comments.

  1. Some sentences are lengthy and might benefit from being broken down for clarity. For example, the sentence starting with "A survey of 17 101 VS patients..." is quite long and could be rephrased for better readability.

Thank you for your helpful comments. I restructured the abstract completely.

  1. The abstract mentions "almost half" of the patients experiencing postoperative headaches, but it might be helpful to provide a specific percentage for a clearer understanding.

Thank you for your helpful comments. I restructured the abstract completely.

  1. While the abstract mentions the use of various surveys and questionnaires, it does not provide details on the methodology used for the study. Including a brief overview of the study design and methods could enhance the abstract.

Thank you for your helpful comments. I restructured the abstract completely.

  1. The abstract discusses associations between postoperative headaches and perceived health benefits, but it does not explicitly mention the direction of causality. Clarifying whether postoperative headaches directly lead to reduced perceived health benefits or if there are other factors at play could strengthen the abstract.

Thank you for your helpful comments. I restructured the abstract completely.

  1. The abstract briefly mentions short-term psychological interventions, but it could benefit from more explicit recommendations for clinical practice or further research based on the study's findings.

Thank you for your helpful comments. I restructured the abstract completely.

  1. It's important to acknowledge any limitations in the study. For example, potential biases in survey responses, generalizability of findings, or other methodological constraints should be briefly mentioned.

Thank you for your helpful comments. I restructured the abstract completely.

  1. The abstract could conclude with a concise summary of the key findings and their potential implications, reinforcing the significance of the study

Thank you for your helpful comments. I restructured the abstract completely.

  1. The introduction covers a range of topics, from the impact of microsurgery on perceived health benefits (PHB) to the association between personality traits and postural recovery. Consider restructuring the introduction to present a clearer flow of ideas, possibly by dividing it into subsections that focus on specific aspects of the study.

Thank you very much for this comment. I restructured the introduction part as follows:

Symptoms Affecting PHB in VS Patients, Premorbid Psychological Factors and POH, Personality Traits in VS Patients, Headaches and Personality Traits, Psychiatric Comorbidities in Headache Patients, Objectives of the Current Study.

  1. While the objectives of the study are mentioned towards the end of the introduction, it might be helpful to explicitly state them earlier for the reader to understand the purpose of the study right from the beginning.

Thank you very much, I added the following passage: “Patients' psychological responses to the surgery, their ability to cope with postopera-tive symptoms, and their overall adjustment to the changes in health status may vary based on their individual personality traits. Exploring these aspects can provide valuable insights into optimizing postoperative care and improving the overall well-being of VS patients undergoing microsurgery.” (ll53-57).

  1. The introduction refers to several previous studies, which is good for contextualizing the research. However, it would be more informative if specific citations were provided for each referenced study, allowing readers to access the relevant literature.

Thank you, I provided citations for the studies.

  1. Some transitions between sentences and ideas could be smoother. Consider using transition phrases to guide the reader through the logical progression of information.

Thank you very much, I added transition sentences, such as “Given that VS patients who underwent retrosigmoid surgery frequently experience POH (13,14), there is a natural progression to exploring associations between headache types and personality traits” (lines 67-69).

  1. The section on personality traits and their impact on postural recovery is informative but could benefit from a brief explanation of how these traits are relevant to the study of vestibular schwannoma (VS) patients.

Thank you, I added further context. I neglected to mention earlier that Ribeyre et al explicitly investigated patients after vestibular schwannoma (VS) surgery.

  1. While it's mentioned that microsurgery might lead to a decrease in PHB compared to conservative therapy, it would be helpful to briefly discuss why this is the case and how it relates to the current study.

Thank you for this valuable comment. I added the following passage to the introduction: “In the current study, understanding the impact of personality traits on coping mechanisms and PHB in VS patients after retrosigmoid microsurgery becomes crucial. Patients' psychological responses to the surgery, their ability to cope with postopera-tive symptoms, and their overall adjustment to the changes in health status may vary based on their individual personality traits. Exploring these aspects can provide valuable insights into optimizing postoperative care and improving the overall well-being of VS patients undergoing microsurgery”.

  1. Ensure consistency in the terminology

Thank you, I updated the terminology throughout the manuscript.

  1. While the descriptive data in Table 1 is useful, consider incorporating more narrative explanation to highlight key findings before directing readers to the tables. For instance, briefly discuss notable trends or patterns in demographic variables before presenting the detailed table.

Thank you, I added ll 168-173.

  1. The description of the study sample includes both "50" and "54" participants with POH, and later in the results section, "50" is consistently used. Clarify the reason for this discrepancy and ensure consistency in

Thank you for your prompt attention to this matter. I believe there may have been a typo on my end. The correct figure is indeed 54, and I have adjusted it accordingly. Your understanding and cooperation are greatly appreciated.

  1. The correlation matrix (Table 3) is extensive and includes various variables. Consider providing a brief narrative interpretation of the most relevant or significant correlations before referring readers to the table for detailed information. This helps readers understand the key relationships without needing to analyze the entire matrix.

Thank you for bringing this to my attention. I've restructured the document to incorporate a narrative summary of the key findings prior to presenting the table. The summarized insights have been organized under relevant subheadings for clarity and coherence. If you have any specific preferences or additional adjustments you'd like, please let me know, and I'll be happy to accommodate.

  1. In Table 2, Cohen's d is provided as a measure of effect size for the Mann-Whitney-U-Tests. Consider providing a brief interpretation or reference point for the Cohen's d values to help readers understand the practical significance of the observed differences.

Thanks for pointing this out, I added tho following sentence: “The effect size of d = 0.52 can be interpreted as medium according to Cohen (33).” (ll 175-176).

  1. Ensure that all abbreviations used in the tables (e.g., NAS, TIPI-G, FESV, HD, ANX, ANG, APC, CR, EC, MD, CA, RR, GBI) are defined either in the text or in a footnote to enhance reader understanding.

Thank you very much, I added the missing abbreviations to the footnote (l 171).

  1. To enhance the structure and readability, consider using subsection headings for different aspects of the results (e.g., "3.1. Demographic Characteristics," "3.2. Associations between Age, Hearing Loss, Facial Paresis, and Pain").

Thanks for pointing this out, I added subsections accordingly.

  1. In the last part of the results, where predictors for perceived health benefits are discussed, provide more interpretation of the findings. Explain the significance of pain-related helplessness and depression and cognitive restructuring as predictors and their implications for perceived health benefits.

Thank you very much, I changed the text under subheading 3.6 Predictors for PHB to “The step-wise regression analysis conducted revealed certain predictors for the GBI total score, which serves as an indicator of PHB. The findings indicate that pain-related helplessness and depression were identified as negative predictors (β = -.34), suggesting that higher levels of pain-related helplessness and depression are as-sociated with lower perceived health benefits. On the other hand, cognitive restructuring emerged as a positive predictor (β = .37), implying that individuals who engage in cognitive restructuring tend to report higher perceived health benefits. The overall model's statistical significance (F (2, 46) = 11.13, p < 0.01, R2 = .33) suggests that these predictors collectively explain a significant portion (33%) of the variability in total GBI scores or perceived health benefits.”

  1. Depending on the complexity of the data, consider incorporating visual aids such as graphs or charts to help readers grasp key trends or relationships more easily.

Thank you for your suggestion. I have incorporated additional figures for correlation analysis and moved Table 3 to the Supplementary section.

  1. The discussion covers a wide range of topics, including associations among personality traits, pain coping, psychological distress, and perceived health benefits (PHB). Consider organizing the discussion into subsections to enhance clarity. For example, have separate sections for personality traits, pain coping, psychological distress, and PHB.

Thank you very much. I added the following subheadings:

4.1 Summary of findings, 4.2 Associations Among Age, Hearing Loss, and Pain-Related Mental Interference, 4.3 Personality Traits and Pain-Related Mental Interference, 4.4 Biopsychosocial Model and Diathesis-Stress Component, 4.5 PHB and Coping Mechanisms, 4.6 Psychological Interventions for POH in VS patients, 4.7 Limitations

  1. While the discussion provides a detailed summary of the findings, consider delving deeper into the potential mechanisms or reasons behind the observed associations. For instance, when discussing the association between personality traits and pain-related mental interference, explore theoretical explanations or existing literature supporting these associations.

Thank you, I added the following passage: “Theoretical explanations for associations between personality and pain coping of-ten center on the psychobiological aspects of pain perception. Personality traits might influence neurobiological processes, including stress response systems and pain modulation pathways. For example, individuals with high extraversion may exhibit more effective stress coping mechanisms, thereby mitigating the impact of stress-induced hyperalgesia. Emotional stability, on the other hand, could contribute to a more adaptive pain appraisal, reducing the emotional amplification of pain signals. Moreover, openness as a personality trait may play a role in shaping cognitive and behavioral responses to pain. Open individuals, characterized by creativity and a willingness to experience novel ideas, may adopt more diverse and flexible coping strategies. This adaptability could extend to their ability to reframe pain-related cognitions, fostering a more positive outlook and mitigating mental interference.” (ll 56-67).

  1. When referring to previous studies (e.g., Garnefski and Kraaij, Turel et al., Magyar et al., Ramírez-Maestre et al.), provide more context and specifics about the methodologies and results of those studies. This will help readers understand the relevance and significance of the current findings in the context of existing literature.

Thanks again for pointing this out. I added context to the studies in the discussion part ll 27-28, 39-41, 45-47, and 57-60.

  1. The limitations section is appropriately included, but it might benefit from further discussion. For example, discuss the potential impact of the sample size on the study's generalizability and the potential biases introduced by the overrepresentation of female patients. Additionally, acknowledge any potential limitations related to the online survey methodology and its implications for data accuracy.

Thank you, I added the following passages: sample size ll 119-126, overrepresentation of female participants ll 129-136 and online survey methodology ll 138-143.

  1. The discussion briefly touches on the clinical implications of the findings, particularly in terms of preoperative patient education and interventions. Elaborate on these implications, discussing how the study results could inform clinical practices, patient care, and potential interventions for patients experiencing postoperative hearing-related issues.

Thank you, I enhanced the passage about psychological interventions (ll 119-135).

  1. The conclusion briefly mentions future research with larger and more balanced samples. Provide more specific recommendations for future research directions, such as exploring additional factors that may influence postoperative outcomes, investigating the effectiveness of specific interventions, or considering longitudinal studies to assess changes over time.

Thank you, I added the following passage: “Conducting longitudinal studies would provide valuable insights into the dynamic nature of postoperative outcomes over time. Tracking patients at multiple points post-surgery can elucidate the trajectory of symptoms, coping mechanisms, and psy-chological well-being, offering a more comprehensive understanding of the long-term impact of surgery on VS patients. Beyond personality traits, researchers could explore additional factors that may influence postoperative outcomes. This may include exam-ining the role of social support, coping strategies, and specific cognitive-behavioral factors. Understanding the interplay between these variables and their collective im-pact on postoperative well-being could guide the development of targeted interven-tions. Lastly, collaborative efforts between neurosurgeons, psychologists, and other healthcare professionals could foster a holistic understanding of postoperative out-comes. Multidisciplinary studies could explore how a combination of medical, psycho-logical, and supportive interventions contributes to better patient outcomes.” (ll 191-203).

  1. Ensure that each statement, especially those making comparisons with previous studies or presenting theoretical concepts, is supported by appropriate citations. This enhances the credibility of the discussion and provides readers with the opportunity to explore relevant literature.

Thank you, I have checked the citations throughout the manuscript.

Reviewer 2 Report

Comments and Suggestions for Authors

The manuscript entitled “ Perceived Health Benefits in Vestibular Schwannoma Patients 2 with Long-term Postoperative Headache: Insights from Person- 3 ality Traits and Pain Coping – a cross-sectional study” is well written and organized.

Some remarks should be taken in considerations “

-        The abstract  :mention clearly how did the survey was applied “ the methods”

-        Lines from 87 – 90 is not well presented and are not informative.

-        What are the information which are missed or not discussed and not well studied previously in the already published articles regarding the aim of the current study? Illustrate this in the aim of the work.

-        How could you adjust the stability of the selected samples?

-        Why the number of “ n “ are different in POH and non POH ?

-        Ensure all the abbreviations were mentioned in the table footnote.

-        Illustration of the table “3” is not well defined in the subsection entitled “ Personality and perceived health benefits”

-        Add the limitations of your study.

Comments on the Quality of English Language

Minor English Editing is required.

Author Response

Dear Reviewer,

thank you very much for taking your time to review our manuscript. We really value your helpful comments.

-        The abstract  :mention clearly how did the survey was applied “ the methods”

Thank you for your helpful comments. I restructured the abstract completely.

-        Lines from 87 – 90 is not well presented and are not informative.

Thank you for your valuable feedback. I have carefully restructured the introduction section in response to the insightful comments provided by Reviewer #1.

-        What are the information which are missed or not discussed and not well studied previously in the already published articles regarding the aim of the current study? Illustrate this in the aim of the work.

Thank you very much, I added the following passage: “Patients' psychological responses to the surgery, their ability to cope with postopera-tive symptoms, and their overall adjustment to the changes in health status may vary based on their individual personality traits. Exploring these aspects can provide valuable insights into optimizing postoperative care and improving the overall well-being of VS patients undergoing microsurgery.” (ll53-57).

-        How could you adjust the stability of the selected samples?

Thank you for this comment. I added this to the discussion part.

-        Why the number of “ n “ are different in POH and non POH ?

Thanks for pointing this out. It was a typo, I changed it.

-        Ensure all the abbreviations were mentioned in the table footnote.

Thank you, I added abbreviations as footnotes.

-        Illustration of the table “3” is not well defined in the subsection entitled “ Personality and perceived health benefits”

Thank you very much. I moved the Table to the Supplements section and added figures instead.

-        Add the limitations of your study.

Thank you very much, I added limitations to the abstract and the discussion.

Round 2

Reviewer 2 Report

Comments and Suggestions for Authors

The authors should pay more attention for the reviewer comments , add the changes in the response letter and give sufficient information about the lines they changed in their revised manuscript. All you have done should be explained in you letter.

The authors, said they have fixed the typos errors in the number of “n” in the table (2), POH and non POH ( 42 and 54 ) , are they are the same?

      Also  How could you adjust the stability of the selected samples?

Thank you for this comment. I added this to the discussion part.:  this should be added in the methods , I can’t find where did you changed.

Comments on the Quality of English Language

Minor Editing 

Author Response

Dear Reviewer,

I sincerely appreciate your insightful comments and constructive feedback. Your guidance has been invaluable in refining the manuscript, and I am grateful for the opportunity to address your concerns.

  1. I have duly considered your suggestion regarding sample sizes and implemented the necessary changes. Specifically, adjustments have been made in Lines 161, 175, and Table 2 to ensure consistency and accuracy in reporting.

  2. Regarding your query about adjusting the stability of the selected samples, I have taken your feedback to heart. Lines 108 to 115 have been added to provide a detailed explanation of the measures undertaken to enhance the stability of the selected samples. I would appreciate further clarification as I want to ensure that my interpretation aligns with your intended meaning. If there was a specific aspect or detail you had in mind that might not have been adequately addressed in my previous response, kindly provide additional guidance or details. 

Furthermore, in response to your recommendations, I have expanded the Limitations section to encompass Lines 371-378, Lines 381-388, and Lines 390-395. These additions provide a more comprehensive acknowledgment of the study's limitations, offering readers a nuanced understanding of the potential constraints and implications of our research.

Once again, I express my gratitude for your time and expertise in reviewing this manuscript. Your input has undoubtedly strengthened the overall quality of the research, and I look forward to any further guidance you may provide.